# Robust TOA-Based UAS Navigation under Model Mismatch in GNSS-Denied Harsh Environments

**Jan Mortier**[ID]**, Gaël Pagès ***[ID]** and Jordi Vilà-Valls**[ID]

Institut Supérieur de l'Aéronautique et de l'Espace (ISAE-SUPAERO), University of Toulouse, 31055 Toulouse, France; jan.mortier@student.isae-supaero.fr (J.M.); jordi.vila-valls@isae-supaero.fr (J.V.-V.)
**\*** Correspondence: gael.pages@isae-supaero.fr

**Abstract:** Global Navigation Satellite Systems (GNSS) is the technology of choice for outdoor positioning purposes but has many limitations when used in safety-critical applications such Intelligent Transportation Systems (ITS) and Unmanned Autonomous Systems (UAS). Namely, its performance clearly degrades in harsh propagation conditions and is not reliable due to possible attacks or interference. Moreover, GNSS signals may not be available in the so-called GNSS-denied environments, such as deep urban canyons or indoors, and standard GNSS architectures do not provide the precision needed in ITS. Among the different alternatives, cellular signals (LTE/5G) may provide coverage in constrained urban environments and Ultra-Wideband (UWB) ranging is a promising solution to achieve high positioning accuracy. The key points impacting any time-of-arrival (TOA)-based navigation system are (i) the transmitters' geometry, (ii) a perfectly known transmitters' position, and (iii) the environment. In this contribution, we analyze the performance loss of alternative TOA-based navigation systems in real-life applications where we may have both transmitters' position mismatch, harsh propagation environments, and GNSS-denied conditions. In addition, we propose new robust filtering methods able to cope with both effects up to a certain extent. Illustrative results in realistic scenarios are provided to support the discussion and show the performance improvement brought by the new methodologies with respect to the state-of-the-art.

**Keywords:** intelligent transportation systems; robust navigation; TOA-based ranging; robust filtering; model mismatch; harsh propagation conditions; outliers

---

## 1. Introduction

Reliable and precise position, navigation, and timing (PNT) information is fundamental in safety-critical applications such as Intelligent Transportation Systems (ITS), automated aircraft landing, or Unmanned Autonomous Systems (UAS), i.e., unmanned autonomous ground/air vehicles (robots/drones). In addition, in the context of ITS, this is not only of capital importance for the autonomously navigating system, but also for vulnerable road users, such as cyclists and pedestrians, and systems collaterally using this information, i.e., in traffic control and for emergency services. Even if the main source of positioning information are still Global Navigation Satellite Systems (GNSS), they lack of reliability and accuracy in constrained environments such as highly populated areas, which limit their adoption as standalone PNT systems. For instance, GNSS may be affected by attacks such as jamming and spoofing [1], or be severely affected in harsh propagation conditions [2]. Moreover, standard GNSS may not provide the precision needed in the ITS context, i.e., submeter lane-level precision. Several GNSS carrier phase-based precise PNT solutions exist to improve the latter, for instance, Real-Time Kinematics (RTK) ([3] Chapter 26) or Precise Point Positioning (PPP) techniques ([3] Chapter 25), but they need either a reference station or precise corrections which do not allow real-time processing, and are even more affected by harsh propagation conditions than standard

GNSS code-based techniques. In addition, these systems may not be available at all in the so-called GNSS-denied environments, such as deep urban canyons or indoors, and therefore robust alternatives must be accounted for.

Several alternatives to GNSS exist: (i) exploitation of cellular signals (LTE/5G) [4] or other signals-of-opportunity (SoO) [5]; (ii) the combination with local inertial navigation systems (INS) [6]; (iii) the use of vehicle-to-vehicle (V2V) or vehicle-to-infrastructure (V2I) communications, such as Dedicated Short-Range Communications (DSRC), which use the IEEE802.11p protocols to obtain peer-to-peer measurements [7,8]; or (iv) more complicated approaches using cameras, LIDARs; and/or radars [9] or in unmanned aerial vehicle (UAV) cooperation scenarios [10,11]. Among these alternatives, extensive research explored the use of cellular signals for positioning (i.e., see [4,12–15] and references therein) and its use as a promising complement to GNSS and other navigation systems [16,17]. However, as pointed out in [17], the combination of both map matching, cellular signals, INS, and GNSS in urban environments, leads to standard deviations approximately equal to 3 m (i.e., $3\sigma$ error around 10 m). Therefore, cellular, GNSS, and standard multisensor data fusion strategies do not provide a precise enough navigation solution for ITS and safety-critical applications such as autonomous driving and UAS. This is even more critical under harsh propagation conditions found in urban environments, which further degrade the performance of all these navigation systems.

In contrast to the use of SoO, such as cellular signals, a different approach is to consider dedicated infrastructure specifically designed to provide precise ranging measurements. This is the case of impulse-radio Ultra-Wideband (UWB) technologies, which are typically based on time-of-arrival (TOA) two-way ranging measurements and can achieve a sub-decimeter level ranging accuracy in line-of-sight (LOS) nominal conditions [5,18], being two orders of magnitude lower than GNSS and LTE ranging measurements. UWB-based positioning has been mainly explored as an alternative for indoor positioning where GNSS is not available [19–23]. With respect to other ranging technologies, UWB has the additional advantage to be more robust to interference.

In general, TOA-based navigation (i.e., UWB or LTE) has the fundamental drawback that transmitters' position is assumed to be perfectly known, which may not be the case in real-life applications. Given the sub-decimeter nominal accuracy of UWB-based systems, this may have a strong impact on the final performance as it has been recently shown in [24]. This is also a critical point if UWB ranging measurements are used in multisensor data fusion platforms, for instance, in combination with GNSS, which require a common navigation coordinate frame [25]. Therefore, in real-life applications, a mismatch on the transmitters' position is a key point to be carefully analyzed for reliable TOA-based navigation. In addition, harsh propagation conditions such as multi-path, fading, and non-line-of-sight (NLOS) conditions may also have a non-negligible impact on the final performance. Then, both model mismatch and harsh environments must be taken into account.

In this contribution, based on the preliminary results in [24], where we proposed an augmented state extended Kalman filter (EKF) to cope with possible UWB anchor position mismatch under nominal Gaussian conditions, we further explore robust TOA-based navigation in realistic scenarios with model mismatch and affected by harsh propagation conditions. We consider that several transmitter to receiver links may be affected by multi-path or NLOS conditions, then we assume the typical contamination model arising in robust statistics [26] for a subset of (corrupted) ranging measurements. Without model mismatch, a possible solution is to consider a robust regression-based EKF [26], but this methodology does not apply when both outliers and model mismatch are present, mainly because the filter cannot distinguish between true measurement outliers and measurements which deviate from the nominal due to the mismatch. Therefore, we propose new robust filtering strategies able to cope with both effects. Illustrative realistic simulation results are provided to support the discussion and show the performance improvement brought by these new robust TOA-based navigation methodologies.

The article is organized as follows. The TOA-based state estimation problem under model mismatch and non-nominal conditions is described in Section 2. State-of-the-art EKF-based solutions

are given in Section 3. The derivation of new robust filtering techniques under model mismatch is detailed in Section 4. The performance of such methods, compared to state-of-the-art solutions, is shown and discussed in Sections 5 and 6. Finally, conclusions and remarks are drawn in Section 7.

## 2. State-Space TOA-Based Navigation Formulation

### 2.1. Nominal State-Space Model

Consider the positioning of a mobile agent (i.e., car, bicycle, pedestrian, or UAS), where at time $t$ both 3D position and velocity are to be inferred, $\mathbf{p}_t = [x_t, y_t, z_t]^T$ and $\mathbf{v}_t = [v_{x,t}, v_{y,t}, v_{z,t}]^T$. If $L$ transmitters (Tx) (i.e., UWB anchor nodes or LTE cellular base stations), within the communication range of the target, are located at fixed 3D positions $\mathbf{p}_i = [x_i, y_i, z_i]^T$, with $i = 1, \ldots, L$, then the measured Tx to agent distance/range is given by

$$z_{i,t} = ||\mathbf{p}_t - \mathbf{p}_i|| + n_{i,t}, \tag{1}$$

where $|| \cdot ||$ is the $L2$-norm,

$$d_{i,t} = ||\mathbf{p}_t - \mathbf{p}_i|| = \sqrt{(x_t - x_i)^2 + (y_t - y_i)^2 + (z_t - z_i)^2},$$

and $n_i$ a measurement noise. The full set of available observations (measurement equation) is written as

$$\underbrace{\begin{bmatrix} z_{1,t} \\ \vdots \\ z_{L,t} \end{bmatrix}}_{\mathbf{z}_t} = \underbrace{\begin{bmatrix} ||\mathbf{p}_t - \mathbf{p}_1|| \\ \vdots \\ ||\mathbf{p}_t - \mathbf{p}_L|| \end{bmatrix}}_{\mathbf{h}_t(\mathbf{x}_t)} + \underbrace{\begin{bmatrix} n_{1,t} \\ \vdots \\ n_{L,t} \end{bmatrix}}_{\mathbf{n}_t}. \tag{2}$$

In the nominal case, $\mathbf{p}_i$ are the true Tx positions and the measurement noise is zero-mean Gaussian, $\mathbf{n}_t \sim \mathcal{N}(\mathbf{0}_L, \mathbf{R}_t)$, with known diagonal covariance $\mathbf{R}_t$ and the individual measurement variance $\sigma_{r,i}^2$ driven by the underlying ranging technology. Notice that in real-life applications the number of available observations may be time-varying (i.e., the number of Tx in view or within the agent communication range), then the system must be aware that the size of the measurement vector may be changing over time.

Regarding the dynamics of the agent, it is common practice, and without loss of generality, to consider a constant velocity model (process or state equation),

$$\underbrace{\begin{bmatrix} \mathbf{p}_t \\ \mathbf{v}_t \end{bmatrix}}_{\mathbf{x}_t} = \underbrace{\begin{bmatrix} \mathbf{I}_3 & \Delta t \mathbf{I}_3 \\ \mathbf{0}_3 & \mathbf{I}_3 \end{bmatrix}}_{\mathbf{F}} \underbrace{\begin{bmatrix} \mathbf{p}_{t-1} \\ \mathbf{v}_{t-1} \end{bmatrix}}_{\mathbf{x}_{t-1}} + \underbrace{\begin{bmatrix} \mathbf{0}_3 \\ \mathbf{w}_{v,t-1} \end{bmatrix}}_{\mathbf{w}_{t-1}}, \tag{3}$$

with $\mathbf{w}_{t-1} \sim \mathcal{N}(\mathbf{0}, \mathbf{Q}_{t-1})$. Both (2) and (3) define the nominal *state-space model (SSM) formulation* of the TOA-based navigation problem, that is, at every time $t$ estimate $\mathbf{x}_t$ from available measurements up to time $t$, $\mathbf{z}_{1:t}$. In the sequel we introduce the concepts of model mismatch and harsh environments.

### 2.2. Non-Nominal Mismatched State-Space Model

To further introduce the main problem to be considered in this contribution, we define the mismatched SSM and how to account for non-nominal propagation conditions.

### 2.2.1. Mismatched SSM

In real-life applications, we may have a partial knowledge or uncertainty in a subset of Tx positions. In this case, a Tx position mismatch, on a subset $\mathcal{U}_e$ of $L_e \leq L$, is written as

$$\tilde{\mathbf{p}}_i = \mathbf{p}_i + \Delta \mathbf{p}_i \text{ for } i \in \mathcal{U}_e, \tag{4}$$

where $\Delta \mathbf{p}_i = [\Delta x_i \ \Delta y_i \ \Delta z_i]^T$ is the position error on the *i*-th Tx, which can be viewed as a bias in its position. The mismatched measured distance (for $i \in \mathcal{U}_e$) is

$$\tilde{z}_{i,t} = ||\mathbf{p}_t - \tilde{\mathbf{p}}_i|| + n_{i,t} = ||\mathbf{p}_t - (\mathbf{p}_i + \Delta \mathbf{p}_i)|| + n_{i,t} = \sqrt{d_{i,t}^2 + \varepsilon_t(\mathbf{p}_t, \Delta \mathbf{p}_i)} + n_{i,t}, \tag{5}$$

with $\varepsilon_t(\mathbf{p}_t, \Delta \mathbf{p}_i) = -2(\mathbf{p}_t - \mathbf{p}_i)^T \Delta \mathbf{p}_i + \Delta \mathbf{p}_i^T \Delta \mathbf{p}_i$, and $\mathbf{p}_t$ and $\Delta \mathbf{p}_i$ (or equivalently $\tilde{\mathbf{p}}_i$) being unknown. We have in this case a new measurement vector $\tilde{\mathbf{z}}_t^T = [\tilde{z}_{1,t}, \dots, \tilde{z}_{L_e,t}, z_{L_e+1,t}, \dots, z_{L,t}]$ (the first $L_e$ distances are affected by the mismatch and the remaining ones are considered to be obtained under nominal known Tx position conditions). The corresponding measurement equation is then

$$\tilde{\mathbf{z}}_t = \tilde{\mathbf{h}}_t(\tilde{\mathbf{x}}_t) + \mathbf{n}_t = \begin{bmatrix} ||\mathbf{p}_t - \tilde{\mathbf{p}}_1|| \\ \vdots \\ ||\mathbf{p}_t - \tilde{\mathbf{p}}_{L_e}|| \\ ||\mathbf{p}_t - \mathbf{p}_{L_e+1}|| \\ \vdots \\ ||\mathbf{p}_t - \mathbf{p}_L|| \end{bmatrix} + \mathbf{n}_t, \tag{6}$$

where $\tilde{\mathbf{x}}_t$ is the augmented state vector of (3) which accounts for the mismatch (refer to Section 3.2).

### 2.2.2. Harsh Propagation Conditions

It is known that, for instance, in urban environments or indoors, there are several propagation effects which deviate from Gaussianity, then being crucial to design robust methods in order to obtain a reliable solutions. A possible way to account for these conditions is to consider parametric non-Gaussian measurement noise distributions, i.e., Student-t, Laplace, or skew-t [27,28], but then the robust filtering solution is constrained to the imposed parametric noise modeling and thus not flexible to time-varying propagation conditions. Another approach is to consider the typical contamination model arising in robust statistics [2,26]: consider a proportion $1 - \varepsilon$ of observations under nominal Gaussian noise, and another proportion $0 \leq \varepsilon \leq 1$ of observations contaminated by an unknown distribution,

$$n_{i,t} \sim (1 - \varepsilon) \mathcal{N}(0, \sigma_{r,i}^2) + \varepsilon \mathcal{H}, \tag{7}$$

where $\mathcal{H}$ is an arbitrary contaminating distribution accounting for possible outliers (i.e., corrupted observations), for instance, a non-zero mean Gaussian distribution with variance $\sigma_H \gg \sigma_{r,i}$.

The main question is *how do we estimate the agent's position and velocity under both Tx position mismatch and non-nominal propagation conditions?*

## 3. State-of-the-Art EKF-Based Solutions

### 3.1. Standard Extended KF Solution

Considering the SSM in (2) and (3), it is easy to design an EKF [29], which needs a linearized (approximated) measurement equation, given by the following Jacobian matrix (evaluated at a given point $\mathbf{x}^0$ and with $\mathbf{u}_{i,t} = ||\mathbf{p}^0 - \mathbf{p}_i||$),

$$\mathbf{H}_t = \left.\frac{\partial \mathbf{h}_t(\mathbf{x}_t)}{\partial \mathbf{x}_t}\right|_{\mathbf{x}^0} = \begin{bmatrix} \frac{x^0-x_1}{\mathbf{u}_{1,t}} & \frac{y^0-y_1}{\mathbf{u}_{1,t}} & \frac{z^0-z_1}{\mathbf{u}_{1,t}} & \mathbf{0}_3^T \\ \vdots & \vdots & \vdots & \vdots \\ \frac{x^0-x_L}{\mathbf{u}_{L,t}} & \frac{y^0-y_L}{\mathbf{u}_{L,t}} & \frac{z^0-z_L}{\mathbf{u}_{L,t}} & \mathbf{0}_3^T \end{bmatrix}. \tag{8}$$

The standard TOA-based navigation solution is obtained by applying the EKF to the assumed models (2) and (3):

$$\hat{\mathbf{x}}_{t|t-1} = \mathbf{F}\hat{\mathbf{x}}_{t-1|t-1}$$
$$\boldsymbol{\Sigma}_{x,t|t-1} = \mathbf{F}\boldsymbol{\Sigma}_{x,t-1|t-1}\mathbf{F} + \mathbf{Q}$$
$$\mathbf{K}_t = \boldsymbol{\Sigma}_{x,t|t-1}\mathbf{H}_t^T(\mathbf{H}_t\boldsymbol{\Sigma}_{x,t|t-1}\mathbf{H}_t^T + \mathbf{R})^{-1}$$
$$\hat{\mathbf{x}}_{t|t} = \hat{\mathbf{x}}_{t|t-1} + \mathbf{K}_t(\mathbf{z}_t - \mathbf{h}_t(\hat{\mathbf{x}}_{t|t-1}))$$
$$\boldsymbol{\Sigma}_{x,t|t} = (\mathbf{I} - \mathbf{K}_t\mathbf{H}_t)\boldsymbol{\Sigma}_{x,t|t-1}$$

with $\mathbf{H}_t = \partial \mathbf{h}_t(\mathbf{x}_t)/\partial \mathbf{x}_t|_{\mathbf{x}_t = \hat{\mathbf{x}}_{t|t-1}}$. Notice that, for $t \geq 1$, the linear KF recursion is only valid if $\boldsymbol{\Sigma}_{x,0|0} = \mathbf{C}_{\mathbf{x}_0}$ and $\hat{\mathbf{x}}_{0|0} = \mathbb{E}[\mathbf{x}_0]$ (mean and covariance of the initial state) [30], which in practice are unknown. An appropriate way to initialize the filter is thus needed.

A practical solution is to use as initial estimate the linear minimum variance distortionless response (LMVDR) estimator, which coincides with the weighted least squares estimator (WLSE) of $\mathbf{x}_1$, and is given by

$$\hat{\mathbf{x}}_{1|1} = \boldsymbol{\Sigma}_{x,1|1}\mathbf{H}_1^T\mathbf{R}^{-1}\mathbf{z}_1, \quad \boldsymbol{\Sigma}_{x,1|1} = \left(\mathbf{H}_1^T\mathbf{R}^{-1}\mathbf{H}_1\right)^{-1}. \tag{9}$$

In this case, due to the nonlinearity of the measurement equation, the WLSE must be replaced by an iterative WLSE.

In case of model mismatch, the standard EKF-based solution is blind because it directly uses the assumed (mismatched) measurements $\tilde{d}_{i,t} = ||\mathbf{p}_t - \tilde{\mathbf{p}}_i||$. This solution can always be used and may provide acceptable results (depending on the application requirements) if $d_{i,t}^2 >> |\varepsilon_t(\mathbf{p}_t, \Delta\mathbf{p}_i)|$.

### 3.2. Augmented-State EKF Solution

An alternative to the previous standard EKF, which does not take into account the possible model mismatch, was recently proposed in [24]. The idea is to consider an augmented state, $\tilde{\mathbf{x}}_t = [\mathbf{p}_t \ \mathbf{v}_t \ \mathbf{p}_{1,t} \ \cdots \ \mathbf{p}_{L_e,t}]^T$, that is, to include the (partially) unknown $\mathbf{p}_i$ into the state to be estimated. The process equation in this case is given by

$$\underbrace{\begin{bmatrix} \mathbf{p}_t \\ \mathbf{v}_t \\ \mathbf{p}_{1,t} \\ \vdots \\ \mathbf{p}_{L_e,t} \end{bmatrix}}_{\tilde{\mathbf{x}}_t} = \underbrace{\begin{bmatrix} \mathbf{I}_3 & \Delta t\mathbf{I}_3 & \\ & \mathbf{I}_3 & \\ & & \mathbf{I}_{3L_e} \end{bmatrix}}_{\tilde{\mathbf{F}}} \tilde{\mathbf{x}}_{t-1} + \underbrace{\begin{bmatrix} \mathbf{0}_3 \\ \mathbf{w}_{v,t-1} \\ \mathbf{0}_3 \\ \vdots \\ \mathbf{0}_3 \end{bmatrix}}_{\tilde{\mathbf{w}}_{t-1}}, \tag{10}$$

and the measurement equation as in (2). The corresponding Jacobian matrix (dimension $L \times (6 + 3L_e)$) is

$$
\tilde{\mathbf{H}}_t\big|_{\tilde{\mathbf{x}}_t = \tilde{\mathbf{x}}^0} =
\left[
\begin{array}{ccc|cccc}
\mathbf{l}_{1,t} & \mathbf{0}_3^T & -\mathbf{l}_{1,t} & & & & \\
\vdots & \vdots & & \ddots & & & \\
\mathbf{l}_{L_e,t} & \mathbf{0}_3^T & & & -\mathbf{l}_{L_e,t} & & \\
\hline
\mathbf{l}_{L_e+1,t} & \mathbf{0}_3^T & \mathbf{0}_3^T & \cdots & & \mathbf{0}_3^T & \\
\vdots & \vdots & \vdots & \ddots & & \vdots & \\
\mathbf{l}_{L,t} & \mathbf{0}_3^T & \mathbf{0}_3^T & \cdots & & \mathbf{0}_3^T &
\end{array}
\right],
\tag{11}
$$

with $\mathbf{l}_{i,t} = \left[ \frac{x - x_i}{\|\mathbf{p} - \mathbf{p}_i\|} \quad \frac{y - y_i}{\|\mathbf{p} - \mathbf{p}_i\|} \quad \frac{z - z_i}{\|\mathbf{p} - \mathbf{p}_i\|} \right]$ being the *i*-th LOS unit vector. It has been shown in [24] that this provides better results w.r.t. the standard EKF if not all the Tx positions are partially unknown (i.e., $L_e < L$). In order to take into account the Tx position uncertainty, the measurement noise covariance $\tilde{\mathbf{R}}_t$ is modified w.r.t. $\mathbf{R}_t$, where the variance of the possibly mismatched measurements is set to

$$
[\tilde{\mathbf{R}}]_{i,i} = \sigma_{r,i}^2 + \sigma_{p,i}^2.
\tag{12}
$$

A possible choice of $\sigma_{p,i}^2$ is to consider a maximum bias, for instance, a uniformly distributed random position bias $\Delta \mathbf{p}_i \in [-\mathbf{b}_i, \mathbf{b}_i]$, with $\mathbf{b}_i^T = [b_{x,i} \ b_{y,i} \ b_{z,i}]$. If the bias is equal on every coordinate, $b_{x,i} = b_{y,i} = b_{z,i} = b_i$, then $\sigma_{p,i}^2 = b_i^2/3$.

However, this solution only tackles the Tx mismatch problem and is not designed, as for the standard EKF, to cope with possible harsh propagation conditions, i.e., non-nominal perturbations on the noise $\mathbf{n}_t$. Therefore, as it will be shown in Sections 5 and 6, it is not a robust solution and new methodologies must be designed for real-life applicability.

## 4. Robust Filtering under Model Mismatch

How to deal with measurements under the contamination model in (7) has been studied for several decades in the field of robust statistics. A recent publication provides an excellent introduction with several applications to practical signal processing problems [26]. In the sequel, we first give a brief introduction on the idea behind the most basic robust methods, and then explain how these concepts can be exploited to design robust filtering methods under model mismatch.

### 4.1. The Idea behind Robust Estimation

We consider as an example the *linear regression problem*, $\mathbf{y} = \mathbf{H}\mathbf{x} + \mathbf{n}$, where we want to estimate $\mathbf{x}$ and the noise components are i.i.d. If we define the residuals $\mathbf{r} = \mathbf{y} - \mathbf{H}\mathbf{x}$ the solution is given by the least squares (LS) estimator,

$$
\hat{\mathbf{x}}_{\mathrm{LS}} = \arg\min_{\mathbf{x}} \|\mathbf{y} - \mathbf{H}\mathbf{x}\|^2 \Rightarrow \arg\min_{\mathbf{x}} \sum_{i=1}^{n} \left( \frac{r_i(\mathbf{x})}{\sigma} \right)^2
\tag{13}
$$

where we introduced the normalization of the residuals. Notice that a single outlier can destroy our estimate; thus, it is not a robust solution. Instead of using a quadratic function of the residuals, we can consider a general loss function $\rho(\cdot)$ (and its derivative for the minimization, $\psi(x) = \frac{\partial \rho(x)}{\partial x}$) as

$$
\hat{\mathbf{x}} = \arg\min_{\mathbf{x}} \sum_{i=1}^{n} \rho\left( \frac{r_i(\mathbf{x})}{\sigma} \right),
\tag{14}
$$

$$
\sum_{i=1}^{n} \psi\left( \frac{r_i(\mathbf{x})}{\sigma} \right) \frac{\partial \left( r_i(\mathbf{x})/\sigma \right)}{\partial \mathbf{x}} = \mathbf{0},
\tag{15}
$$

and, for instance, $\rho_{LS}(x) = x^2$ and $\rho_{\ell_1}(x) = |x|$ correspond to the LS and $\ell_1$ estimators. The most common $\rho(\cdot)$ is the so-called Huber function:

$$\rho(x) = \begin{cases} x^2 & \text{if} \quad |x| \leq a \\ 2a|x| - a^2 & \text{if} \quad |x| > a \end{cases}, \tag{16a}$$

$$\psi(x) = \begin{cases} x & \text{if} \quad |x| \leq a \\ a \, \text{sign}(x) & \text{if} \quad |x| > a \end{cases}, \tag{16b}$$

$$w(x) = \begin{cases} \psi(x)/x, & \text{if } x \neq 0 \\ \psi'(0), & \text{if } x = 0 \end{cases} = \min\left\{1, \frac{a}{|x|}\right\}, \tag{16c}$$

where the value $a$ is chosen to obtain a given efficiency (deviation from the optimal under nominal conditions), for instance, $a = 1.345$ for a 95% efficiency. The idea is that residuals with large errors are downweighted. To solve the problem in (15) we need the residuals standard deviation $\sigma$, or a robust estimate of it, for instance the normalized median absolute deviation (MAD), $\hat{\sigma}_M$, defined as

$$\hat{\sigma}_M(\mathbf{x}) = c_m \, \text{Med}(|\mathbf{x} - \text{Med}(\mathbf{x})|), \tag{17}$$

where $\text{Med}(\mathbf{x})$ is the median and $c_m$ a normalizing constant (typically $c_m = 1.4815$). Considering the weight function in (16c) it is easy to see that (15) can be rewritten as

$$\sum_{i=1}^{n} w(r_i/\hat{\sigma}_M) \, \frac{r_i}{\hat{\sigma}_M} \, \frac{\partial(r_i/\hat{\sigma}_M)}{\partial \mathbf{x}} = 0, \tag{18}$$

which can be solved by an iterative reweighted LS and is the so-called regression M-estimator. Notice that other cost functions $\rho(\cdot)$ can be considered, and more evolved estimators such as the MM-estimator exist [26]. As discussed in [2], for TOA-based navigation problems an important point is the measurement redundancy, that is, having enough observations to be able to find the solution.

### 4.2. Standard Robust Regression EKF

The robust regression M-estimator introduced in Section 4.1 can be used within the EKF framework in order to obtain a robust EKF [26]. Notice that the problem of interest is the one where we may have outliers in the observations $\mathbf{z}_t$ and the residuals are related to the innovation sequence $\mathbf{z}_t - \mathbf{h}_t(\hat{\mathbf{x}}_{t|t-1})$. A first approach is to use a robust score function $\psi(\cdot)$ in order to directly downweight the innovation vector, then not modifying the EKF recursion. Another more general approach is to reformulate the EKF as a regression problem and then use at every $t$ a robust regression M-estimate of $\mathbf{x}_t$, being the so-called robust regression EKF (RREKF) ([26] Chapter 7). Notice that both approaches were not developed to cope with a possible model mismatch and they are not directly suited for the navigation problem in this contribution (i.e., as it will be shown in Section 6). Obviously, one can combine the original RREKF with the augmented state in Section 3.2, but the underlying problem to apply these techniques to the mismatched SSM is that the filter is not able to distinguish between true measurement outliers and measurements which deviate from the nominal due to the mismatch. Therefore, new robust solutions under mismatch must be sought.

### 4.3. Robust Regression EKF for Mismatched Models

Carefully analyzing the robust M-estimator downweighting process, that is, $\rho\left(\frac{r_i(\mathbf{x})}{\hat{\sigma}_M}\right)$, we see that the normalizing term $\hat{\sigma}_M$ is a key parameter for the filter. Indeed, if $\hat{\sigma}_M$ is larger than the nominal variance, large outliers will not be downweighted and degrade the solution. In contrast, if this term is lower than the nominal then meaningful information will be disregarded by the filter. Recall that the problem is to deal with possibly corrupted measurements where the noise is distributed according

to the contamination model in (7). In addition, some of the measurements may be biased due to the model mismatch. Then, in practice, we can modify (7) to include the effect of the mismatch as

$$
\begin{aligned}
n_{i,t} &\sim \mathcal{N}(0, \sigma_{r,i}^2), & \text{prob } (1-\varepsilon), & \quad \text{no mismatch} \\
n_{i,t} &\sim \mathcal{N}(0, \sigma_{r,i}^2 + \sigma_{p,i}^2), & \text{prob } (1-\varepsilon), & \quad \text{mismatch} \\
n_{i,t} &\sim \mathcal{H}, & \text{prob } \varepsilon, & \quad \text{outliers}
\end{aligned}
$$

With the nominal distribution being $\mathcal{N}(0, \sigma_{r,i}^2)$, the main problem of the RREKF is that it is not able to correctly distinguish among $\mathcal{N}(0, \sigma_{r,i}^2 + \sigma_{p,i}^2)$ and $\mathcal{H}$. Moreover, within the augmented state EKF framework, measurements arising from $\mathcal{N}(0, \sigma_{r,i}^2 + \sigma_{p,i}^2)$ should not be understood as faulty measurements, because these observations are the ones that allow the filter (which includes $[\mathbf{p}_{1,t} \ \cdots \ \mathbf{p}_{L_e,t}]^T$ in the state) to converge to the true Tx positions, and recursively correct the mismatch.

Then, a possible robust EKF solution which is able to cope with model mismatch (named MRREKF), and thus avoid to downweight measurement errors induced by model mismatch is

(i)     first, consider the augmented-state formulation in Section 3.2 to cope with the Tx mismatch;
(ii)    second, consider a RREKF-based solution; and
(iii)   within the M-estimator, instead of the MAD, use a mismatched three sigma rule to normalize the residuals, that is,

$$
\hat{\sigma}_{3s} = 3\sqrt{\sigma_{r,i}^2 + \sigma_{p,i}^2}. \tag{19}
$$

This may be seen as a compromise between the $3\sigma$-rejection or score function type robust EKF ([26] 7.3.1) and the robust regression KF ([26] 7.3.5), in order to control the filter behavior under model mismatch. Surprisingly, to the best of our knowledge, this simple solution has not been explored in the literature. As it will be shown in Section 6, this method behaves extremely well for the problem under study. The price to be paid w.r.t. the standard and augmented state EKFs, is that this solution includes an iterative procedure to find the robust estimate at every Kalman iteration. The iteration procedure terminates either after a fixed number of iterations or when the normalized difference between the actual and previous state estimate is smaller than a fixed threshold. Thus, the computational time will mainly depend on these two parameters. This may be critical in very low cost platforms and real-time applications. This is the reason why we propose an low cost alternative in the sequel.

*4.4. Computationally Efficient Solution: Robust Weighting Uncertainty Indicator for Robust EKF*

One of the limitations of the previous RREKF and MRREKF is their computational complexity, that is, within every EKF time step there is an iterative reweighted LS algorithm to solve the regression M-estimation problem, which, depending on the application at hand, may be a limiting factor. Therefore, we consider another computationally light alternative to the RREKF, which relies on the use of a robust weighting function in order to update the statistical characteristic of the measurement noise variance. Indeed, the performance of the EKF is highly affected by the measurement covariance matrix, $\mathbf{R}_t$; therefore, one or more contaminated measurements will have a significant impact on the filter performance.

The proposed method is based on the values taken by the normalized residuals (i.e., innovations) as introduced in Section 4.1. In order to reduce the impact of the contaminated measurements on the estimator, the idea is to increase the variance of each contaminated measurement depending on the value of $|r_i/\hat{\sigma}_M|$. To do so, we propose in this study to use the square inverse of Huber's weighting function, as shown in Figure 1, to adjust the measurement noise variances. Thus, in the case of large residuals, i.e., $|r_i/\hat{\sigma}_M| > a$, and in the framework of Tx position mismatch, the variance can be formulated as

$$[\tilde{\mathbf{R}}]^*_{i,i} = \frac{1}{(w\,(r_i/\hat{\sigma}_M))^2}\,[\tilde{\mathbf{R}}]_{i,i}. \tag{20}$$

Replacing $w\,(r_i/\hat{\sigma}_M)$ by (16c), the latter can be rewritten as

$$[\tilde{\mathbf{R}}]^*_{i,i} = r_i^2\,\frac{[\tilde{\mathbf{R}}]_{i,i}}{a^2\hat{\sigma}_M^2}, \tag{21}$$

where we can see that the variance associated with a contaminated measurement is proportional to the square of the normalized residual. In other words, the larger the outlier is, the larger is the associated variance, which in turn implies a direct downweight within the EKF through the Kalman gain. Therefore, with a minor EKF modification, the filter is able to use the robust weight as an outlier uncertainty indicator, which at the end does not impact the augmented-state EKF behavior to correctly deal with model mismatch. This new robust covariance weighting method is subsequently denoted MRCEKF.

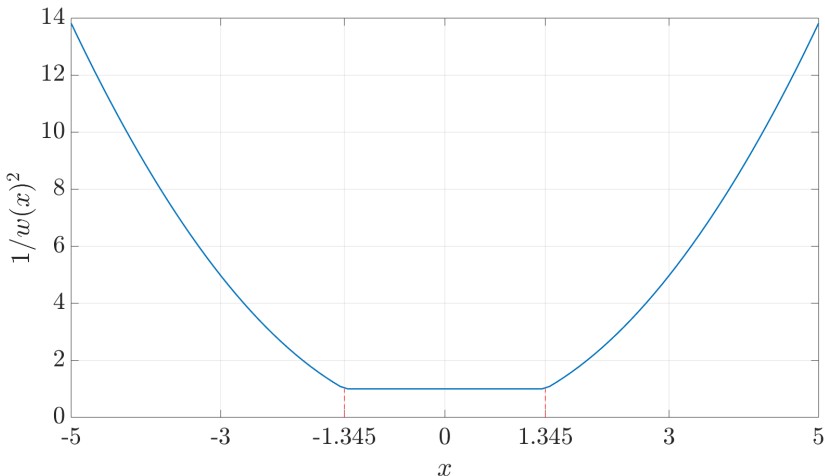

**Figure 1.** Square inverse of Huber's weighting function.

## 5. Illustrative UWB-Based Indoor Navigation Example

The performance of the new robust filtering methods in Sections 4.3 and 4.4 is first assessed in an indoor environment where only UWB measurements are available. The set of methods considered is as follows.

- Standard EKF (SEKF) in Section 3.1.
- Augmented-state EKF (MEKF) in Section 3.2.
- Robust regression EKF (RREKF) from ([26] Chapter 7).
- Robust regression EKF for mismatched models (MRREKF) in Section 4.3.
- Robust covariance weighting EKF (MRCEKF) in Section 4.4.

The Huber's score function ($a = 1.345$), as defined in Section 4, is used to compute the RREKF, MRCEKF, and MRREKF state estimates. The results are obtained from 100 Monte Carlo runs.

### 5.1. Simulation Scenario

We consider $L = 8$ anchors and a realistic mobile agent trajectory, shown in Figure 2. The position of five anchors (A4–A8) is considered to have a 3-dimensional bias which is drawn randomly in $[-0.5, 0.5]$ m, for each Monte Carlo run. The unknown state vector to be estimated is then $\mathbf{x}_t = [\mathbf{p}_t, \mathbf{v}_t, \tilde{\mathbf{p}}_4, \dots, \tilde{\mathbf{p}}_8]^T$. The initial position and velocity of the agent were set to $\mathbf{p}_{m,0|0} = (0, -2, 0.5)^T$ and $\mathbf{v}_{m,0|0} = (0, 0, 0)^T$ plus a random initial bias of $\pm0.5$ m and $\pm0.01$ m $\cdot$ s$^{-1}$, respectively. The noise of

the inlier observations is described by a zero-mean Gaussian distribution whose variance was set to $\sigma_{r_i,\text{LOS}}^2 = 0.01$ m$^2$. Similarly, the multipath/NLOS effect (outliers) in the observations is modeled as a zero-mean Gaussian distribution with a variance $\alpha^2$ times larger than that of the inlier observations. The mixed LOS/NLOS conditions are modeled as a Markovian process that switches between the LOS and multipath/NLOS conditions. Different percentages, $\varepsilon$, for the the probability of being in multipath/NLOS conditions, and outlier magnitudes, $\alpha$, are considered and indicated in Table 1. The number of anchors in multipath/NLOS conditions is drawn randomly within the 8 anchors, and the number of regions and length of each region in mixed LOS/NLOS conditions are also randomly generated to correctly analyze the statistical behavior.

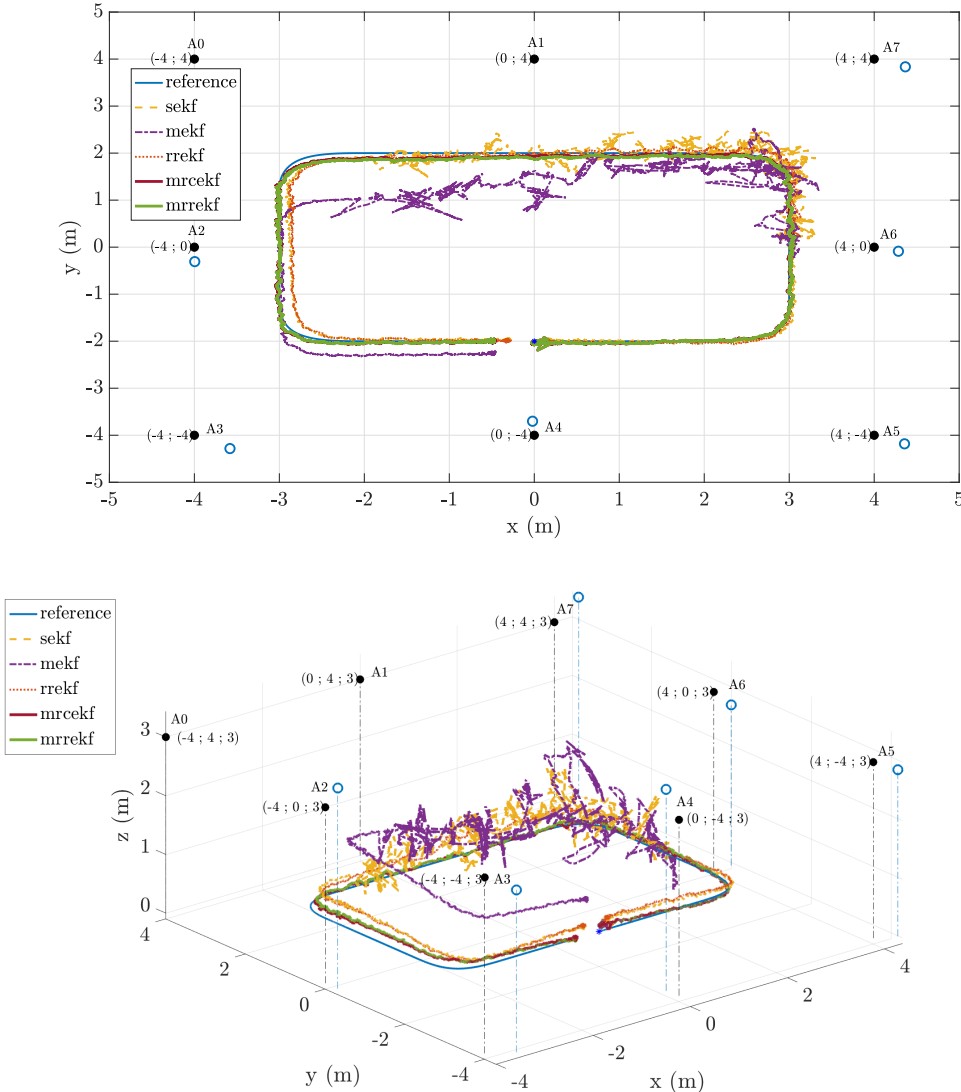

**Figure 2.** 2D (**top**) and 3D (**bottom**) trajectories and the corresponding estimates. The scenario is based on 5 anchors out of 8 having a random position bias of $\pm0.5$ m (blue circles). The mixed line-of-sight/non-line-of-sight (LOS/NLOS) conditions are generated for a specific region of the trajectory and concerns all 8 anchors. They are modeled as a Markovian process with the following parameters, $\sigma_{r_i,\text{NLOS}} = 3$ m, $\varepsilon = 25\%$.

**Table 1.** Parameters for the Monte Carlo simulation.

| | |
|---|---|
| Number of runs | 100 |
| Range variance noise (m$^2$) | 0.01 |
| Number of anchors | 8 |
| Number of mismatched anchors | $L_e = 5$ |
| Anchor position bias (cm) | $[-50, +50]$ |
| Outlier percentage $\varepsilon$ | 10–25–50 |
| Outlier magnitude $\alpha$ | 1–5–10–30–60 |
| Number of multipath/NLOS anchors | 2–4–6–8 |

*5.2. Results*

The MATLAB simulations were run on a Linux Ubuntu 18.04 OS-based laptop with an Intel$^{\circledR}$ Core$^{\text{TM}}$ i5 processor at 1.7 GHz and 16 Go RAM. First, we show in Figure 2 a single realization, and the corresponding five estimates, to clearly illustrate both the impact of the mismatched anchors (blue circles) and harsh propagation conditions (outliers), which are evident in such results. The former can be seen in the 3D plot where the standard SEKF and RREKF are not able to deal with the position mismatch and are affected by a misalignment of the estimated trajectory. Even if the RREKF correctly discards outliers (see 2D plot) it trusts the mismatched anchor positions as being the real ones. The SEKF is not able to cope with neither the mismatch nor outliers. Regarding the augmented state techniques, the MEKF is able to deal with the mismatch but is severely affected by outliers which are understood as Tx position errors. Notice that only the new MRCEKF and MRREKF provide a smooth trajectory, properly dealing with both position mismatch and corrupted measurements.

To give an idea of the order of magnitude in the computational complexity increase of the MRREKF with respect to the MRCEKF (which has no nested loops), the computational time was evaluated using MATLAB's profiler, for the whole simulation. Setting the iteration break to 25 and the threshold to $10^{-4}$ for the MRREKF (and RREKF), it turns out that the total computational time for the MRREKF is of 14.688 s against 4.640 s for the MRCEKF. Further analysis is needed if one wishes to implement the algorithms on a real-time embedded system, which we leave for future work.

Figure 3 shows the average horizontal and vertical root mean square errors (RMSE) obtained with the five estimators as a function of the multipath/NLOS standard deviation (i.e., $\sigma_{r_i,\text{NLOS}} = \alpha\sigma_{r_i,\text{LOS}}$). We provide the results obtained for two, four, six, and eight anchors under multipath/NLOS conditions, and $\varepsilon = 10$, 25 and 50%.

- As expected, for a given percentage of contamination, if the outliers are not very strong ($\alpha \leq 1$ m), the MEKF performs much better than the SEKF, but it breaks down rapidly as the amplitude of the outliers increases. Indeed, the MEKF tries to compensate the modeling error due to the mismatch, but the measurements, which are corrupted by outliers, are understood by the filter as a bias in the anchors' position. In comparison, the SEKF is less affected by this behavior, with no clear performance breakdown, but in general this method provides a worse estimate when compared to the two robust estimators.
- W.r.t. the MEKF behavior, the standard RREKF estimator is relatively stable, thus correctly dealing with outliers, but its nominal performance is degraded because of the mismatch, therefore not bringing a suitable solution for the problem at hand. For low outlier magnitudes, $\alpha$, it is clear, when comparing the outcome of both MEKF and RREKF, that a method being able to cope with both mismatch and outliers should retain the best qualities of both estimators.
- Indeed, the compromise between the MEKF and RREKF is brought by the new MRCEKF and MRREKF, which are relatively stable and cope perfectly with contaminated observations and anchor position mismatch, up to a certain level of contamination. Notice that the interest of the MRREKF is clear with a large outlier amplitude. The MRCEKF outperforms the rest when the number of contaminated measurements is less or equal than 6 and $\varepsilon \leq 50\%$, and the MRREKF provides a good solution up to eight contaminated anchors and for $\varepsilon = 50\%$. It can be concluded,

from the analysis of these results, that for $\varepsilon < 50\%$, $\alpha \leq 3$ m and the number of anchors in multipath/NLOS conditions $\leq 6$, the MRCEKF approach guarantees horizontal and vertical performances below the tens of centimeters, unlike the other standard approaches. For more extreme propagation conditions, the MRREKF is needed and we can not avoid the iterative M-estimate procedure within every Kalman filter iteration.

These indoor navigation results (i.e., standalone UWB positioning) confirm that both new MRCEKF and MRREKF are promising approaches to provide a high-precision solution in challenging propagation conditions for real-life safety-critical applications. To further support this statement, in the sequel we analyze a realistic outdoor urban environment.

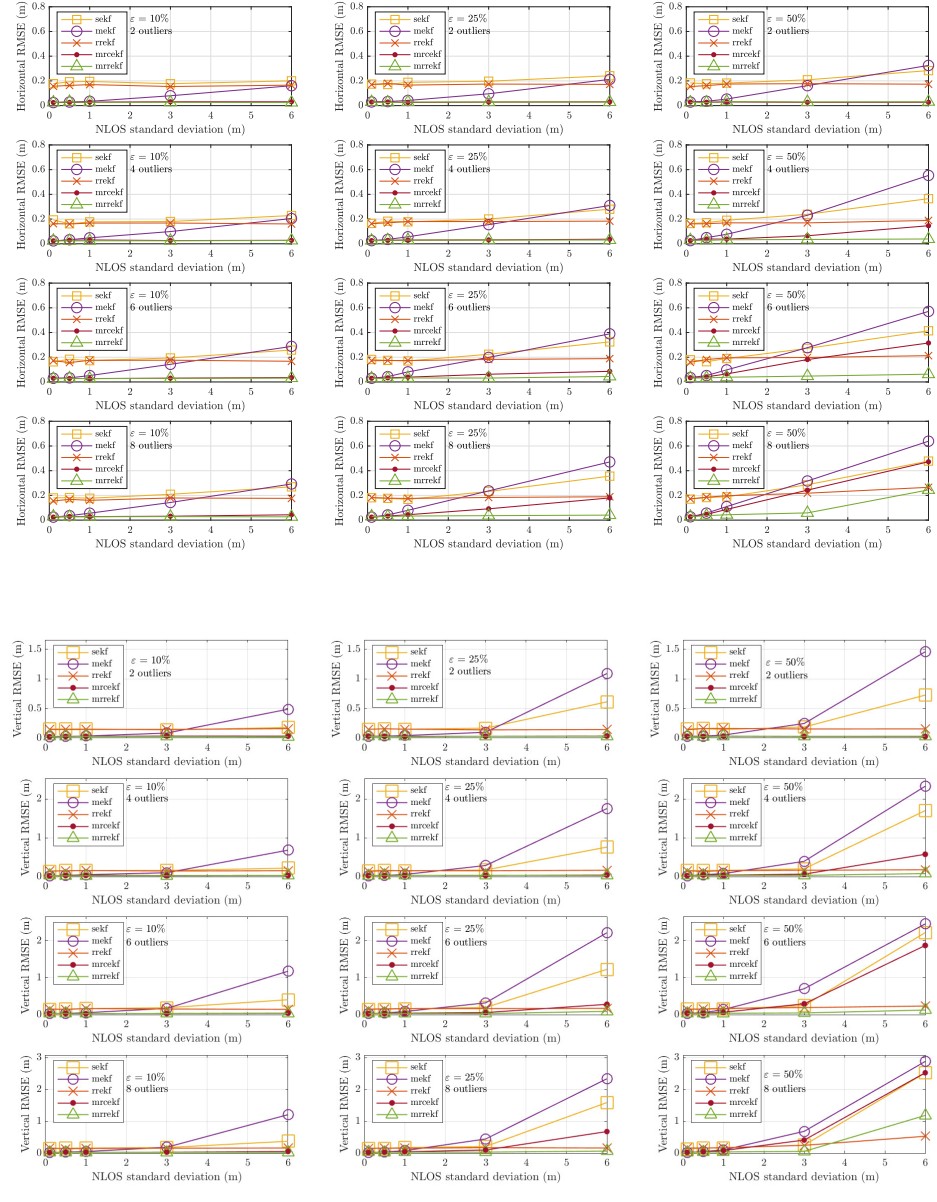

**Figure 3.** Average horizontal (**top**) and vertical (**bottom**) root mean sqaure error (RMSE) results using the standard extended Kalman filter (SEKF), the mismatch extended Kalman filter (MEKF), the robust regression extended Kalman filter (RREKF), and the mismatch robust weight-based extended Kalman filter (MRCEKF).

## 6. Realistic UWB/LTE TOA-Based Urban Navigation Example

In this section, we want to confront the performance of the SEKF with the new MRCEKF and MRREKF, considering a realistic urban environment, with both UWB and LTE measurements available. In order to correctly assess the behavior of the proposed methods, the choice was to use realistic synthetic data instead of a real measurement campaign. It is known that the latter may lead to unexpected behaviors in the hardware or the data acquisition process, out of the scope of this contribution. Therefore, in order to characterize the filtering methods we resort to a randomized Monte Carlo analysis, which is not environment-dependent, and which may account for all possible situations encountered in real-life environments (i.e., tunnels, tree-lined streets, or tall buildings) by inducing different types of noise, outliers, filter initialization, etc.

The scenario implies the use of a *mobile agent*, i.e., a low-speed ground vehicle that will follow a simulated path composed of four narrow streets. The algorithms operate in a 3D state-space, so they are also valid for low-speed aerial vehicles such as drones. High-speed vehicles are out of the scope of this contribution. The trajectory was generated using a realistic trajectory generator in which the model is based on a constant velocity model. Smooth transitions occur between segments, which use a continuous acceleration profile.

### 6.1. Realistic Urban Scenario Definition

We have chosen a realistic urban scenario in downtown Toulouse, France, where the buildings are tall enough and the streets narrow enough to be in severe urban canyon propagation conditions. We consider that a total of 32 mismatched UWB anchors are placed on four-meter-high street lamps (alternatively on both sides of the street) all along a 4-street trajectory. This scenario is chosen because of the presence of 3 LTE antennas located in the surroundings, installed on the roof-top of buildings, which are 3 fixed and perfectly known anchors (LTE eNodeB type in which characteristics are defined in [17]). In a real-life scenario, an RTK-based solution could be used to determine the positions of the LTE base stations with a centimeter-level accuracy. The origin, i.e., (0,0,0), of the East-North-Up (ENU) coordinate frame is defined by the LTE base station number 1. The other two base station coordinates, as well as the UWB anchors, are then expressed in this ENU frame. Refer to Figures 4–6 for details.

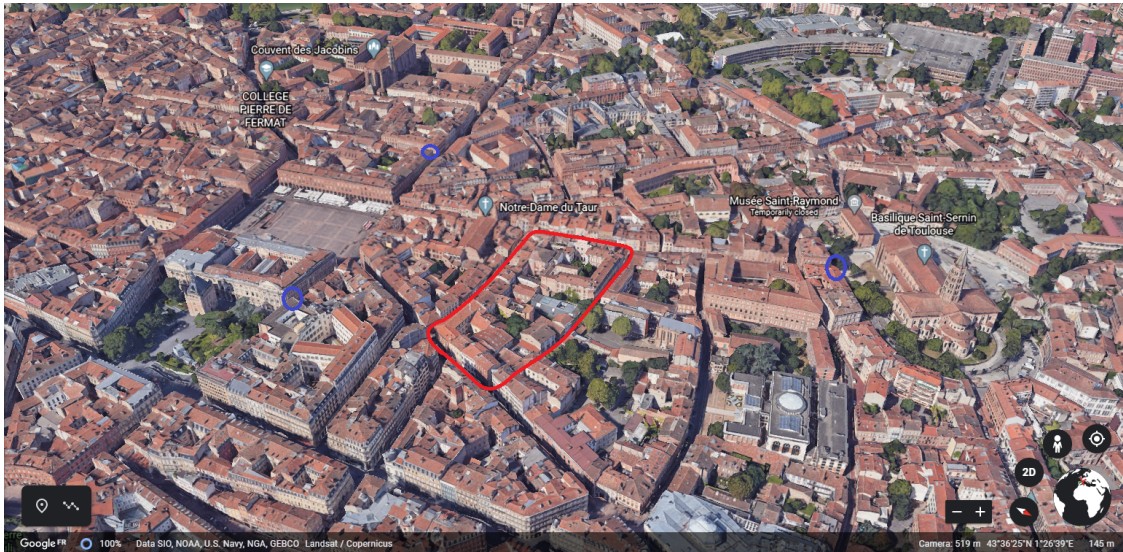

**Figure 4.** 3D Image of the Urban Scenario with the mobile agent's real trajectory (red), and the three LTE antennas (blue circles).

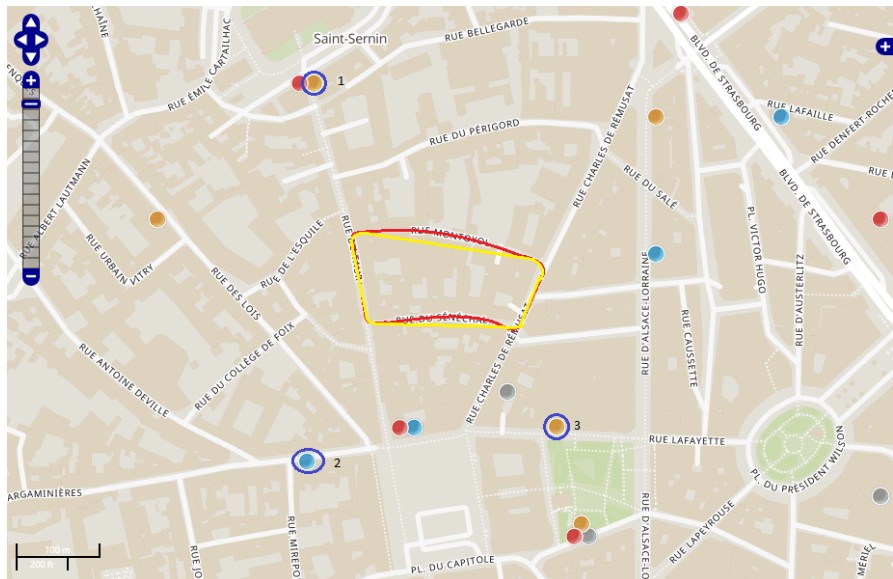

**Figure 5.** 2D map of the urban scenario with the agent's real trajectory (red), the simulated trajectory (yellow), and the LTE antennas with their labels (blue circles).

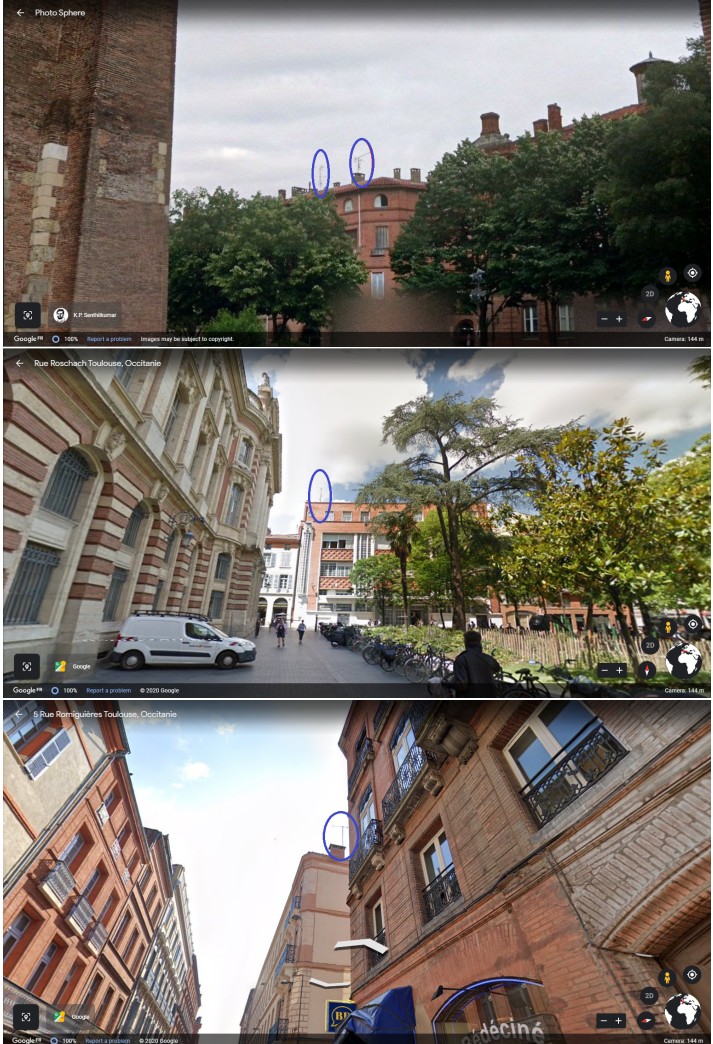

**Figure 6.** Images of LTE antennas, each one located on a building's roof, circled in blue.

We consider 32 identical UWB anchors, that is, they have the same statistical properties in terms of Gaussian noise standard deviation, measurement range bias, maximum position bias, and multipath/NLOS noise amplitude. Their true position is precisely known in order to obtain the ranges and trajectory ground truths, which will not be available to the user (indeed, we recall that all 32 UWB anchors have a position mismatch). The mobile agent follows the 4-street trajectory counterclockwise and spends 50 s in each street. A street is defined by the trajectory between two consecutive corners. We arbitrarily choose the mobile agent's starting point, with a random initial position bias of $\pm 3$ m and zero initial velocity with a random initial bias of $\pm 0.01$ m·s$^{-1}$. Because of the urban canyon nature of this scenario, the mobile agent does not have a communication link available to all UWB anchors, but only to the 7 nearest ones: the *reference anchor*, i.e., the closest to the agent; 3 upstream anchors; and 3 downstream anchors. Considering the 3 LTE base stations, the mobile agent has a total of 10 signals available at each time step.

An example of this situation is depicted on Figure 7, where we can see the real trajectory of the mobile agent (in blue), the three LTE antennas with their labels, and the 32 UWB anchors represented by the black spots near the trajectory. As we use 3 LTE antennas, the UWB anchors will have labels from 4 to 35. The starting point of the mobile agent is shown by the blue cross near anchor 4. In this situation, the agent is the red spot located near anchor 12, which is its current *reference anchor*, thus the 7 UWB anchors having a communication link with the mobile agent will be anchors 9 to 15.

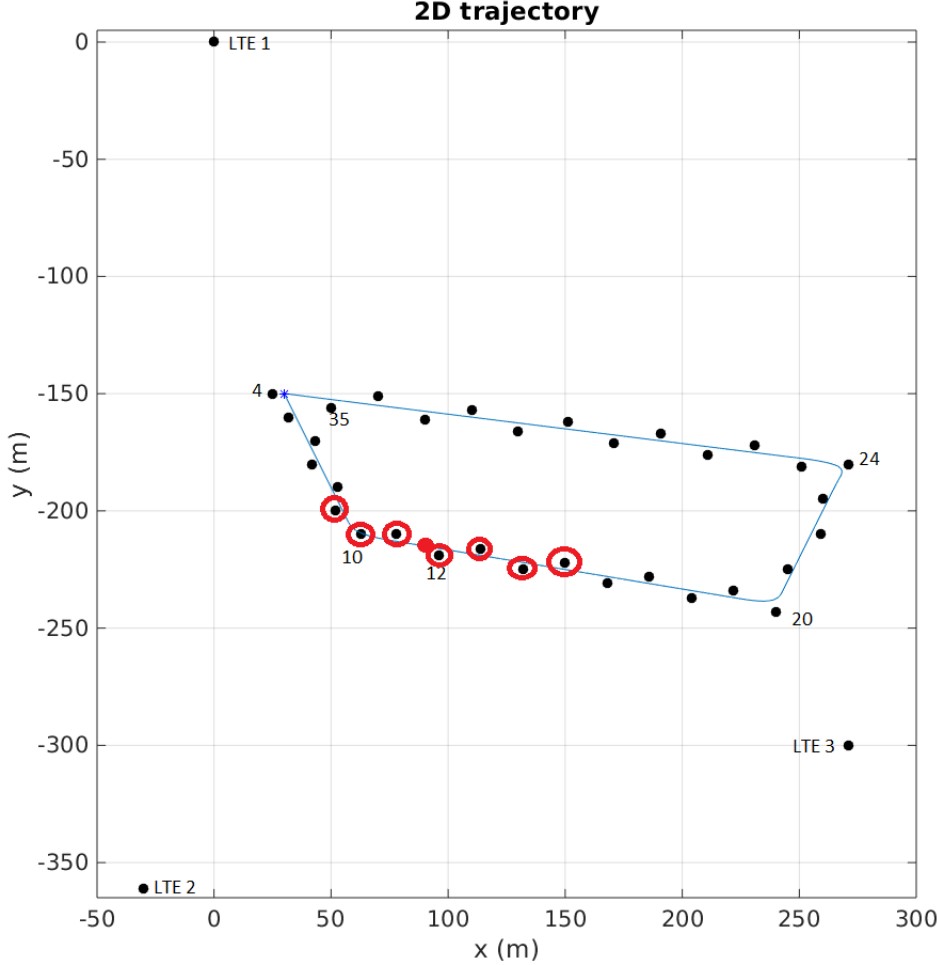

**Figure 7.** Example of the number of communications links with the mobile agent located near anchor 12. The anchors having a link are circled in red.

Moreover, we also consider that the percentage of outliers affecting the different signals available is not the same for all visible anchors, and changes depending on the mobile agent's location. We therefore make the following simulation hypothesis.

- Anchors located in the agent's *reference anchor* street have an outlier percentage of $\varepsilon = 10$, that is, mild multipath probability due to the narrow streets set-up.
- Anchors located in a corner of the trajectory have an outlier percentage of $\varepsilon = 25$, that is, a larger multipath conditions probability.
- The rest of the anchors not on the same agent's *reference anchor* street have an outlier percentage of $\varepsilon = 50$, induced by the lack of direct visibility, i.e., NLOS conditions.

Because the percentage of outliers changes during the agent's trajectory every time it switches *reference anchor*, we will neither study the influence of the number of mismatched anchors, nor the number of anchors under multipath/NLOS conditions, but rather the influence of the anchor position bias and the outlier magnitude $\alpha$. We consider anchors' maximum position bias varying from 1 m to 10 m, and $\alpha$ changing from 30 to 300, which means a multipath/NLOS standard deviation from 3 m to 30 m. The set-up is summarized in Table 2.

**Table 2.** Parameters for the Monte Carlo simulation—Urban canyon scenario.

| | |
|---|---|
| Number of runs | 1000 |
| UWB range variance noise (m$^2$) | 0.01 |
| LTE range variance noise (m$^2$) | 30.25 |
| Number of anchors | 35 (3 LTE + 32 UWB) |
| Number of mismatched anchors | $L_e = 32$ |
| Anchor position bias (m) | $[-1, +1], [-3, +3], [-5, +5], [-10, +10]$ |
| Outlier magnitude $\alpha$ | 30–50–100–150–200–300 |

*6.2. Results*

The average horizontal and vertical RMSE have been evaluated for each estimator over 1000 Monte Carlo runs, as shown in Figures 8 and 9. To obtain a representation of any urban environment, random anchors' position bias, noise, and different types of outliers (Table 2) were generated at each Monte Carlo run.

At first sight, the SEKF and RREKF provide similar results as in Section 5, i.e., they are not able to cope neither with the position bias nor with outliers, whereas the new robust algorithms better deal with both effects, providing a horizontal RMSE which is almost always sub-metric, no matter the position bias or the outlier magnitude. However, when we focus on the curves' trend, we can notice that increasing the position bias tends to flatten the curves, as if the algorithms were more sensitive to the anchors' position bias than to the outlier magnitude, which means that the robustness to harsh propagation conditions is always effective. For instance, the MRREKF is more responsive to the multipath/NLOS standard deviation when $\Delta \mathbf{p}_i \in [-1, +1]$ (the curve is gently increasing with $\alpha$). In comparison, when the maximum position bias increases to 10 m, the curve is almost flat and the horizontal RMSE is sub-metric whatever the multipath/NLOS magnitude. In that case, the algorithm struggles to deal with the high position bias and is not impacted by the fact that the magnitude of outliers is more and more important.

Concerning the vertical RMSE, all algorithms show meter-level performances in most of the cases. Indeed, the vertical position estimate suffers from large uncertainties due to the poor vertical diversity of the UWB anchors, and the LTE base stations as well. As pointed out in the introduction, the network geometry has an impact on the minimum achievable performance, but such analysis is out of the scope of this contribution. Nevertheless, these results once again depict the performance improvement provided by the two new robust algorithms, which are the MRCEKF and the MRREKF.

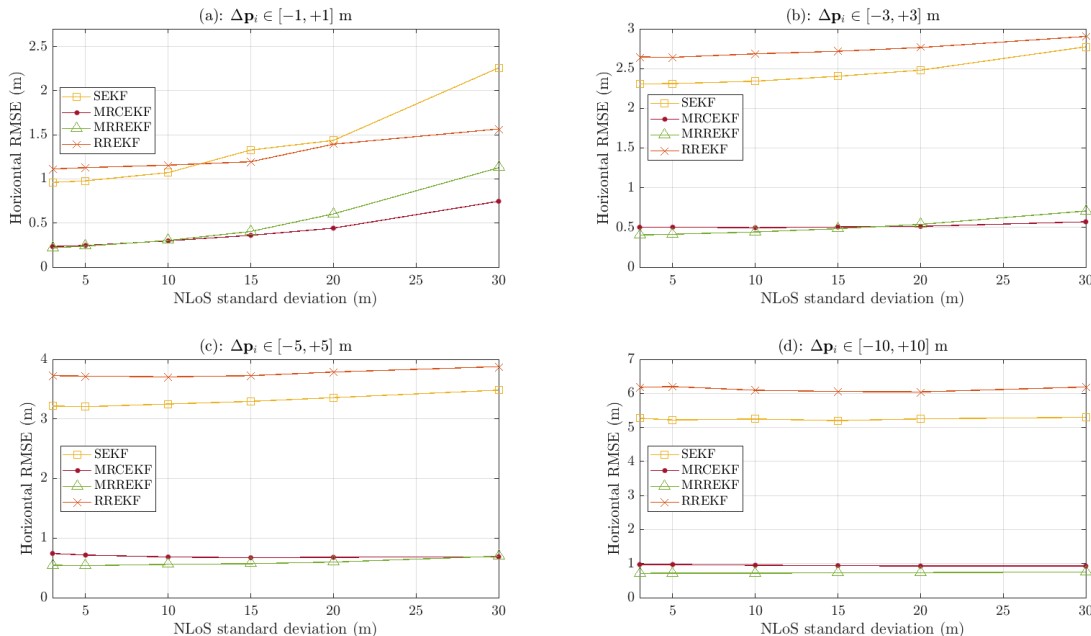

**Figure 8.** Horizontal RMSE results of the Monte Carlo simulation on 1000 runs. Each plot depicts a different anchor position bias $\Delta \mathbf{p}_i \in$ (**a**) $[-1, +1]$ m, (**b**) $[-3, +3]$ m, (**c**) $[-5, +5]$ m and (**d**) $[-10, +10]$ m.

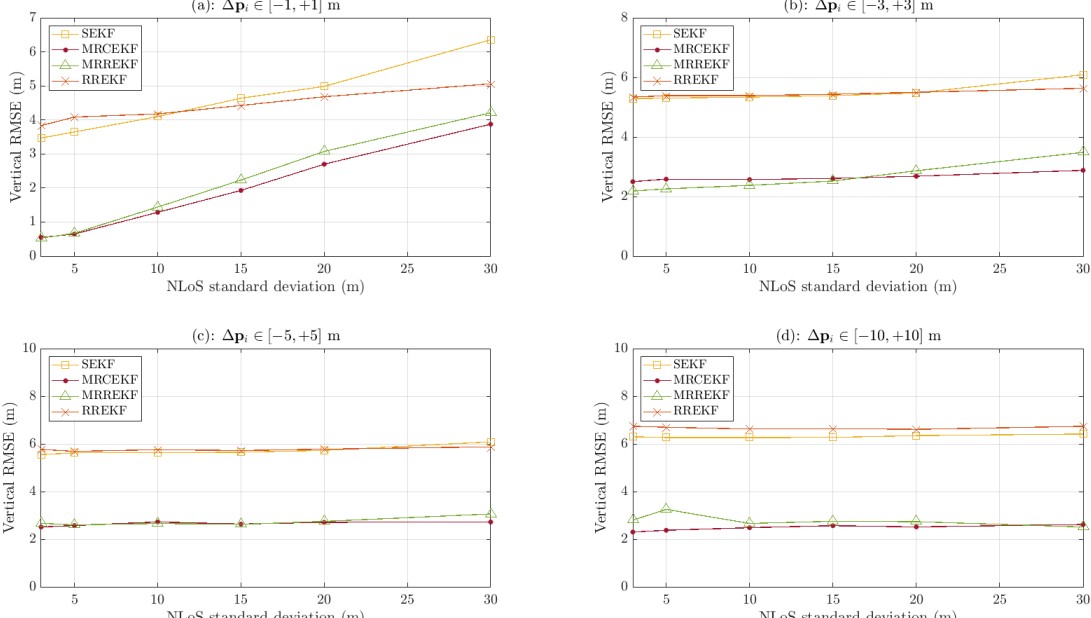

**Figure 9.** Vertical RMSE results of the Monte Carlo simulation on 1000 runs. Each plot depicts a different anchor position bias $\Delta \mathbf{p}_i \in$ (**a**) $[-1, +1]$ m, (**b**) $[-3, +3]$ m, (**c**) $[-5, +5]$ m and (**d**) $[-10, +10]$ m.

## 7. Conclusions and Outlook

In this contribution, we derived two new robust filtering techniques able to deal with both model mismatch (i.e., transmitters' position uncertainty) and harsh propagation conditions (i.e., measurements corrupted by outliers). Such methodologies rely on an augmented state, which includes the uncertain anchors' position, and tools arising from robust statistics. It was also shown that standard techniques suffer from a non-negligible performance loss if mismatch and outliers are not properly taken into consideration; therefore, the new methodologies are promising solutions for real-life safety-critical applications. The first methodology is based on the standard robust regression M-estimation EKF, with the penalty of needing an iterative procedure within every KF step, which may

not be a good option for very low cost platforms. The second methodology focused on the latter, to reduce the computational complexity, by leveraging a robust weighting function to adapt the measurement noise statistics. Overall, both methods provide a significant performance improvement under several configurations with respect to the state-of-the-art.

Note that the vehicle model used in this contribution is a low-speed ground vehicle, which could correspond to a car moving through small and narrow streets in a city center. Notice that the algorithms operate in a 3D state-space, so they are also valid for low-speed aerial vehicles such as drones. High-speed vehicles are out of the scope of the contribution, and indeed, even if the KFs can also be used in this case in such environments, they are not likely to have strong outliers (open-sky environments or rural areas). Therefore, even if the new filtering approaches are valid for any type of vehicle, dynamics, and propagation conditions, their clear benefit is in situations where GNSS is not available and the propagation conditions are severe. Notice that the focus of this contribution was on GNSS-denied harsh environments such as indoors and deep urban canyons, but the new methodologies proposed in the article can also be applied in the context of GNSS and multisensor data fusion (where other types of mismatch may arise), setting the basis for a robust multisensor navigation framework to be further explored.

Another issue worth mentioning is time synchronization. First, notice that the standard way to exploit UWB for positioning is through a two-way ranging protocol within the UWB network. This avoids the need to synchronize the UWB Tx/Rx. In addition, there are several ways to synchronize or time-stamp LTE measurements together with UWB measurements, for instance, using the NTP networking protocol for clock synchronization. One could also imagine that a GNSS receiver is also available but not used for positioning due to its very degraded performance. Notice that even in very harsh propagation conditions, a GNSS receiver provides a clock synchronization in the order of 100 ns, which is more than enough to avoid worrying about the LTE data synchronization. Therefore, the proposed approach without taking into account the LTE clock errors is valid to statistically characterize the new filtering methods, which are intended to deal with model mismatch and outliers.

As a short-term perspective, these techniques must be assessed with real-data experiments, which is currently under study because other problems related to the hardware and the specific platform appear. In the long-term, such techniques could be embedded in a complete GNSS/UWB/LTE/IMU/Lidar/Vision platform to deal with a full set of heterogeneous measurements and model mismatches.

**Author Contributions:** Conceptualization, G.P. and J.V.-V.; methodology, G.P. and J.V.-V.; software, J.M. and G.P.; validation, J.M., G.P., and J.V.-V.; writing—original draft preparation, J.M., G.P., and J.V.-V.; writing—review and editing, J.M., G.P., and J.V.-V.; funding acquisition, G.P. and J.V.-V. All authors have read and agreed to the published version of the manuscript.

**Funding:** This research was supported by the DGA/AID (French Secretary Of Defense) projects (2019.65.0068.00.470.75.01, 2018.60.0072.00.470.75.01).

**Conflicts of Interest:** The authors declare no conflicts of interest.

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
