# Peer review of "Robust TOA-Based UAS Navigation under Model Mismatch in GNSS-Denied Harsh Environments"

_remotesensing, doi:10.3390/rs12182928_

Round 1

Reviewer 1 Report

This work presents an improvement in fusing data from different sources to have a better pose estimation. It is inspired to solve the issue of harsh and urban canyons. The addition of the information of the LTE signal is interest and can reduce the GNSS error. The authors also estimate the bias removing. It is also propositioning a new EFK to solve the GNSS pose estimation issues.

However, the data used to evaluate the new approach is very limited. Figure 7 is not well presented. It is expected to have at least the scale of the grid.

It looks the test was performed in a very small loop with a single loop closure. Nowadays, when it is referend to pose estimation, is required to have long data acquisition with several loop closure. Besides, it needs samples of hazard environments, such as tunnels, tree-lined streets, the city downtown with higher buildings.

It is expected a better evaluation when a new algorithm is proposed.

Reviewer 2 Report

In this paper, the authors present a navigation system for autonomous agents
aimed at handling both transmiters's position mismatch and measurements
corrupted by outliers. Filtering methods are developed for this purpose.
This reviewer recommends that the following minor issues be addressed.
The following statement in the abstract is confusing: "...GNSS-denied harsh
propagation conditions, i.e., measurements corrupted by outliers..." It
should be clarified whether the approach addresses situation when GNSS is
absent or when it is only affected by occasional large errors. Other related
formulations throughout the text must be revised accordingly.
The simulation scenario must be described with more details. The mobile agent
appears to be a low speed ground vehicle, but in Figures 4 and 5 the
trajectories seem to go over buildings. Some details on the simulation model
would also be helpful.
A discussion should be included regarding the limitations of the approach.
This must be performed in terms of vehicle nature (e.g. ground, aerial), speed,
and parameters related to filters and the severity of sensor malfunctions. If
numerical values are available or can be reasonably obtained, that would be
great, but at least a qualitative discussion would be valuable.
The list of references appears to ignore the very numerous publications on
handling GNSS-denied navigation of unmanned aerial vehicles, including the use
of EKF and derived solutions. Unless this reviewer is missing any, there is no
reference to publications in AIAA journals or conferences. To put the proposed
method into context completely, its benefits when compared to such existing
methods should be emphasized.

Reviewer 3 Report

This paper mainly talks about robust UAS navigation system under model mismatch and GNSS-denied environments. And the new algorithm has been validated by the simulation scenario and realistic urban scenario. Here is my comments for this paper. (1) The section 4.3 MRREKF is important for this paper, it seems that only the residuals are normalized compared with the augmented-state EKF, the innovation should be improved and presented. (2) For section 4.4 MRCEKF, you should add a flowchart to make it more complete. And also, you should compare the computation efficient of your algorithms in the test. (3) In the real urban scenario test, how to determine the reference? And why the LTE range is considered though the base position is perfectly known as the range variance noise is high? (4) What’s the sampling rate of UWB? And the test description should be improved. (5) In Figure 8, you list some Monte Carlo simulation results. Could you show the real positioning results during the realistic urban scenario test? (6) The format and the word size of figures should be improved according to the Remote Sensing guideline. (7) The vertical RMSE can also be added in Figure 8. And the analysis for the figure 8 should be more specific. The improvement of two new robust algorithms should be presented compared with the traditional RREKF.

Reviewer 4 Report

The paper presented a robust filter method for GNSS-denied harsh environment. But there exists following major problems:

  1. The clock biases of receiver and transmitters are not considerate in the paper. Actually, this is a main error in such GNSS-denied applications. When such clock errors are introduced as another n-states, it can not easily be distinguished by errors of transmitters' position. In another word, what is the difference between clock error and position error of transmitter. The clocks of LTE station and UWB are not atomic one. So, these terms should not be ignored.
  2. When the errors of transmitters' positions are introduced as states, there are no new measurements introduced (different as authors said near eq. (6)). Thus, it will decrease the observability and cause divergence of the filter. The problems should be analyzed in the paper.
  3. Can this filter be applied in the static application? If there is no motion, how can these position mismatches be estimated. Also, the initial position is also challenge.

The minor problems:

  1. In line 118, using p_1,t and p_le,t to represent the errors of transmitters' positions are confusing with p_m,t. Another symbols should be used.
  2. In line 120, why the position errors pi are uniformly random distributed?

Round 2

Reviewer 4 Report

After reading the feedback from the authors, there are still some comments for the revision:

  1. The clock biases of receiver and transmitters are still not considerate in the paper. The authors proposed following solutions: a) UWB sync. b) NTP protocol. c) GNSS clock. But is all UWB transmitter sync? Is NTP protocol accurate (ms level) enough? Is using GNSS appropriate for GNSS-denied environment? And 100ns clock error means 30m error in position. Also, I looked detail on the ION paper. It said, "The GNSS is used to provide precise time-stamping to the other sensor measurements (IMU and UWB)" (page 7) and "UWB measurements are synchronized with the GNSS clock" (page 6). So, using a state model without consider the clock error is not good solution.
  2. If there are at least 3 well known transmitter's positions, is it still necessary to use the other uncertain transmitters? Will these augmented-states really improve the performance? In practice, how can we make sure we have at least 3 well known transmitters in all trajectory.
  3. The initial value setting of KF filter is critical, due to the estimation of the Tx's position error (augmented-states) depend on it. How can you sure that the initial position provided by the iterative reweighted LS is good enough to let the EKF-based solutions converge? To my opinion, the initial position error is related the Tx's position mismatch. How can we calculate one without consider another?

Due to the concerns in previous cycle are not clear answered, I think the paper need further modification.
